# Functional Analysis of Maize *SDG102* Gene in Response to *Setosphaeria turcica*

**DOI:** 10.3390/plants14223463

**Published:** 2025-11-13

**Authors:** Xin Qi, Xing Zhang, Xiaoxiao Ma, Xinyi Zhao, Xinyang Liu, Xiaoshuang Wei, Huai Tian, Yang Liu, Jianhua Zhang, Zhenhui Wang

**Affiliations:** 1Faculty of Agronomy, Jilin Agricultural University, Changchun 130118, China; xinq@jlau.edu.cn (X.Q.); zhangxing19991115@163.com (X.Z.); 18844065830@163.com (X.Z.); liuxinyang1219@163.com (X.L.); weixiaoshuang@jlau.edu.cn (X.W.); th1990599739@gmail.com (H.T.); liuyuxi5389@163.com (Y.L.); 2Changchun Quality and Safety Testing Center of Agricultural Products, Changchun 130033, China; maxiaoxiao91@163.com

**Keywords:** DEGs, northern corn leaf blight, RNA-seq, disease resistance, *Zea mays*

## Abstract

Northern corn leaf blight (NCLB), caused by the fungal pathogen *Setosphaeria turcica*, is a devastating foliar disease that significantly threatens maize production in China. Previous studies have demonstrated that *SET domain gene 102* (*SDG 102*), a gene encoding an H3K36 methyltransferase, plays a crucial role in regulating maize growth, development, and stress responses. This study used the wild-type (WT), *SDG102* overexpression line (OE), and silencing line (SL) of the corn inbred line B73 as materials. After artificial inoculation with *S. turcica*, the phenotypic characteristics, disease index, yield, and other related traits of different strains were compared, and RNA-Seq was used to analyze the changes in the gene expression profile. The results showed that overexpression of *SDG102* significantly inhibited pathogen spore germination and hyphal growth and enhanced the activity of antioxidant enzymes and the ability to scavenge reactive oxygen species in plants prior to *S. turcica* infection, the opposite trend was observed in *SDG102* silencing lines. Compared with the wild-type, 1546 and 1837 differentially expressed genes (DEGs) responsive to *S. turcica* were identified in OE and SL, respectively. These differentially expressed genes primarily function in pathways such as plant–pathogen interactions, plant hormone signaling, and secondary metabolite biosynthesis. In the OE lines, genes related to plant–pathogen interactions, reactive oxygen species (ROS) production, and key phenylpropanoid biosynthesis genes exhibited higher expression levels. Furthermore, *SDG102* regulates the synthesis of auxin (JA) and abscisic acid (SA) as well as the transcription of their signaling pathway genes, thereby influencing maize resistance to large leaf spot disease. Under corn leaf blight conditions, *SDG102* overexpression increased yield by 9.29% compared to WT, while SL reduced yield by 10.10%. In conclusion, *SDG102* enhances maize resistance to NCLB by positively regulating the expression of disease resistance genes, antioxidant enzyme activity, and hormone-mediated defense pathways.

## 1. Introduction

Northern corn leaf blight (NCLB), caused by the ascomycete fungus *S. turcica*, is a severe foliar disease in maize [1]. The most severe yield losses occur when infection takes place before silking and spreads to upper leaves during the grain-filling stage [2]. Over 4.5 billion people worldwide rely on maize for at least 30% of their food calories [3]. The worsening effects of plant disease outbreaks jeopardize maize production and have far-reaching consequences for food security. Resistant cultivars are the most effective, economical, and environmentally friendly means to control NCLB [4]. Both dominant genes (*Ht1*, *Ht2*, *Ht3*, *HtN*, *Htm1*, *Htn1*, and *HtP*) and recessive genes (*ht4* and *rt*) conferring resistance to specific races of *S. turcica* have been identified [5,6], and several *Ht* genes have been mapped with molecular markers. The first NCLB resistance gene, *Ht1*, was identified in the 1960s, and was widely deployed in commercial hybrids due to its ability to confer a distinct chlorotic resistance response until the emergence of virulent *S. turcica* races that overcame *Ht1*-mediated resistance [5]. Subsequently, additional resistance genes, including *Htn1 (HtN)*, *Ht2*, and *Ht3*, were reported in 1975, 1977, and 1981, respectively. These genes induce similar chlorotic lesions but with significantly reduced necrosis compared to *Ht1* [5]. Given that pathogenicity in *S. turcica* is race-specific, monitoring the distribution of physiological races in maize-growing regions is critical for effective NCLB management.

In recent years, increasing lines of evidence show that epigenetic factors act, as key regulators involved in the transcriptional reprogramming, to modulate plant immune responses [7]. Histones represent a class of evolutionarily conserved [8], modulating chromatin architecture and compaction states to regulate gene transcription and expression patterns [9,10]. Histone methylation is mediated by histone lysine methyltransferases (HKMTases). Typically, all known plant histone lysine methyltransferases (HKMTs) contain an evolutionarily conserved SET domain. Plant histone lysine demethylases (HKDMs) comprise two families: lysine-specific demethylase homologues and Jumonji-C domain-containing proteins. So far, a number of HKMTs and HKDMs have been reported as involved in plant responses to pathogens [11]. In Arabidopsis, H3K36 methylation is catalyzed by the histone methyltransferases *SDG8* and *SDG26*, which are involved in regulating many plant growth and developmental processes [12,13]. *SDG8* is the major HKMT catalyzing H3K36me2/3 [14]. Epigenetics plays a crucial role in regulating nuclear processes in fungal plant pathogens [15], *SDG8* plays a vital role in plant defense against fungal pathogens by modulating a range of genes in JA and/or ET signaling pathways. Loss-of-function of *SDG8* results in mutant plants exhibiting decreased resistance to the fungal pathogens Alternaria brassicicola and Botrytis cinerea [16] as well as to the bacteria pathogen Pst DC3000 [17]. It was proposed that *SDG8*-mediated histone H3K36 methylation may act as a memory mechanism helping them to achieve rapid transcription upon fungal infection or JA treatment [16]. The histone methyltransferases *SDG708*, *SDG724* and *SDG725* are responsible for H3K36 methylation in rice, and loss of each protein causes a late-flowering phenotype [13,18]. *SDG725* is also involved in brassinosteroid-regulated pathways and position-specific intron retention [19,20] and *SDG708* may be involved in stress responses. In rice, JMJ705 specifically removes H3K27me2/3 and its overexpression results in the preferential activation of biotic stress response genes marked by H3K27me3, which enhances rice resistance to bacterial blight disease pathogen Xoo [21]. In addition to *SDG8*, *SDG5* may also play role in plant immunity [22]. Studies by Lee and co-authors indicate that in A. thaliana, *SDG8* and *SDG25* contribute to plant resistance to B. cinerea. *SDG8* appears to perform the deposition of H3K36me2 and H3K36me3, while *SDG25* may deposit H3K4me1 [23]. Despite these advances, the role of H3K36 methylation in maize resistance stress remains unexplored, warranting investigation into its species-specific regulatory mechanisms.

The most cost-effective method for controlling NCLB at present is to create resistant cultivars. *SDG102* is the closest maize homolog of both rice *SDG708* and Arabidopsis *SDG26* [24]. Plants deficient in *SDG102* exhibited more DNA damage than WT plants after UVB irradiation [25]. However, there have been no reports regarding the functions and roles of SDG proteins in maize disease tolerance and development due to the lack of mutants with obvious phenotypes. In order to methodically examine the functional processes of *SDG102* in response to *S. turcica* infection, this study used the maize inbred line B73 wild-type in conjunction with *SDG102* silencing lines and overexpression lines. Phenotypic characterization at the flare-opening stage post-inoculation, illness index evaluation, physiological and metabolic profiling, and transcriptome sequencing (RNA-Seq) were among the thorough assessments. Our research offers theoretical underpinnings for disease resistance breeding techniques based on epigenetics and fundamental insights into the role of histone methylation changes in maize defense responses against *S. turcica*.

## 2. Results

### 2.1. The Effect of SDG102 Gene to Maize Resistance Against S. turcica

In order to examine the function of *SDG102* in defense responses against *S. turcica* infection, we used wild-type, *SDG102*-overexpressing, and *SDG102*-silenced lines of maize inbred B73. Disease progression was tracked at 0, 5, 10, and 20 days post-inoculation (dpi). Neither OE nor SL leaves showed any obvious disease lesions at 5 dpi. All genotypes displayed disease symptoms by 10 dpi, lesion areas remained below 5% of the total leaf area. While OE plants showed noticeably fewer signs of disease at 20 dpi, SL plants showed noticeably larger lesion regions and more lesions than WT (Figure 1). According to these findings, silencing *SDG102* promotes susceptibility to the necrotrophic fungus *S. turcica*, whereas overexpressing it boosts resistance.

We used aniline blue staining to observe the growth process of *S. turcica* on leaves of different materials. At 5 days, conidia and hyphae were clearly visible on the leaf surfaces of both wild-type and silenced lines. At 10 days, hyphae were visible within epidermal cells surrounding the infection site, exhibiting significantly increased elongation compared to day 5 and initiating invasion of the leaf epidermis. OE leaves showed distinct, blue-stained hyphae on day 10, with subsequent hyphal growth progressing more slowly than in WT and SL. By day 20, SL mesophyll cells were nearly completely blocked by mycelial growth, with hyphae spreading into adjacent cells causing cellular damage. Conidiophores formed through stomata. Morphologically, lesions began to expand and merge, leading to leaf wilting (Figure 2). Results indicate that OE leaves inhibit *S. turcica* spore germination and mycelial growth.

### 2.2. Effects of S. turcica Infection on Physiological Parameters of Maize

With OE displaying the highest levels at 5 dpi and SL peaking at 10 dpi, the soluble sugar content first rose and then fell. By 20 dpi, there were notable drops in all genotypes. Over the course of infection, the soluble protein concentration gradually dropped, with OE exhibiting rather steady levels in contrast to SL lines. Super oxide anion (O_2_^−^) content in both transgenic lines changed in two phases, while hydrogen peroxide (H_2_O_2_) buildup peaked at 10 dpi, with larger levels seen in OE plants. Compared with WT, the MDA content in SL leaves was significantly higher than that in the control group at 10 days post-inoculation with *S. turcica*, with increases of 55.30% and 56.90% in SL and OE leaves, respectively. Overall, as the infection period extended, OE plants demonstrated stronger ROS scavenging capacity than the wild type and were able to mobilize a faster pathogen immune response (Figure 3a).

The CAT activity in OE leaves was significantly higher than that in the control, peaking at 20 days post-infection. During the late stage of pathogen infection, the POD activity in SL leaves consistently exceeded that in WT and OE, with a more pronounced increase. In the early stage of infection, APX activity in SL was significantly lower than in OE and SL. The PPO activity in the OE line was significantly higher than that in the SL line, while the activity in the SL line was significantly lower than that in the wild-type. PPO activity in both transgenic lines peaked at 5 days post-infection and subsequently declined to lower levels by day 20. PAL activity exhibited an initial increase followed by a decrease. Compared to WT, PAL activity in OE and SL leaves showed significant changes 5 days after *S. turcica* infection. OE and SL exhibited increases of 9.30% and 8.5%, respectively. Second, PAL enzyme activity in OE leaves peaked at 10 days post-*S. turcica* infection (Figure 3b).

### 2.3. SDG102’s Resistance to NCLB

The expression of the *SDG102* gene under *S. turcica* stress showed an early increase followed by a drop, starting to rise at 5 dpi, peaking at 10 dpi, and drastically declining by 20 dpi (Figure 4b). We also looked at the expression of genes linked to antioxidant enzymes and resistance to northern corn leaf blight. The genes related to resistance to NCBL, such as *ZmPR4*, *ZmPP2C26*, *ChSK1*, *ZmNPR1* and *ZmERF061* were found to be upregulated in OE lines and downregulated in SL lines (Figure 4c), respectively. The relative expression levels of antioxidant enzyme-related genes *ZmPOD*, *ZmSOD*, *ZmPAL*, *ZmCAT*, and *ZmGPX1* were significantly higher in OE than in WT, significant downregulation in SL lines (Figure 4d).

### 2.4. Analysis of Differentially Expressed Genes in Response to S. turcica Infection and KEGG Classification Annotation

While SL compared WT showed 641 upregulated and 937 downregulated DEGs at 5 dpi, OE and SL shared 261 commonly upregulated and 389 generally downregulated DEGs. At 5 dpi, OE versus WT showed 1738 upregulated and 730 downregulated DEGs. In comparison to WT, OE showed 1070 upregulated and 476 downregulated DEGs at 10 dpi, whereas SL showed 902 upregulated and 935 down-regulated DEGs. Between the two transgenic lines, there were 1110 generally upregulated and 869 commonly downregulated DEGs. Significantly, at both time points, OE retained 443 consistently upregulated and 171 consistently downregulated DEGs, while SL displayed 143 and 165 persistent DEGs, respectively (Figure 5a). This suggests that *SDG102*-mediated responses to *S. turcica* have different temporal expression patterns.

KEGG functional annotation revealed that differentially expressed genes in the OE strain primarily participated in pathways including Plant–pathogen interaction, Plant hormone signal transduction, MAPK signaling pathway-plant, Starch and sucrose metabolism, and Phenylpropanoid biosynthesis (Figure 5b). Similar functional annotation results were observed in SL lines (Figure 5c). Under *S. turcica* infection, OE and SL may resist *S. turcica* stress by regulating similar biological processes.

### 2.5. Analysis of SDG102-Mediated Regulation of Plant–Pathogen Interaction Pathway Genes

Among the three lines, the expression of genes involved in the plant–pathogen interaction pathway varied significantly differences (Figure 6a). With expression levels significantly greater than in wild-type plants, we found 103 DEGs in OE lines (Figure 6b), primarily containing WRKY transcription factors, CPK, and CCR. Interestingly, a number of genes that code for PR proteins showed quite substantial overexpression (Figure 6c). Seventeen of the 69 pathway-related DEGs found by comparative analysis of SL lines exhibited significant differential expression with distinct functional interpretations (Figure 6d). Under *S. turcica* infection, defense-related DEGs were significantly enriched in OE compared to SL, indicating that *SDG102* likely functions as a transcriptional regulatory mechanism to modulate maize disease resistance by altering gene expression patterns during plant–pathogen interactions.

### 2.6. SDG102 Modulates Hormone-Mediated Defense Responses Against S. turcica Infection

Under *S. turcica* infestans stress, 31 DEGs that jointly participate in plant hormone signal transduction were identified. Compared to WT, 77 unique DEGs were identified in OE, while 57 unique DEGs were identified in SL (Figure 7a) include ERF1B, CTR, MYB family transcription factors, IV.1, SD2-5, PR, and others. Analysis and comparison of the transcriptional levels of these genes revealed that, compared to the wild-type (WT), the expression levels of these genes in the over-expressing (OE) line were generally higher than those in the wild-type (Appendix A). Comparative transcriptome analysis showed notable changes in SA-, JA-, and ETH-related signaling genes (Figure 7b–d). We quantitatively assessed the overall levels of SA and JA in leaves. Transgenic and WT plants did not significantly differ in their SA or JA con-tent under normal circumstances. Both hormones, however, accumulated considerably after pathogen inoculation; overexpression lines had significantly larger levels of SA and JA than WT, whereas silenced lines had significantly lower levels of JA than controls (Figure 7e,f). We used qRT-PCR to examine the expression patterns of important marker genes in the JA (*JAZ1*, *MYC2*) and SA signaling pathways in order to better describe these responses. While SL lines displayed reduced responses, OE lines quickly saw an increase in JA signaling components (*COI1*, *MYC2*). Interestingly, several SA signaling components (e.g., NPR1) showed downregulation solely in WT/OE contrasts, although the SA downstream marker PR1 was considerably increased in both WT/OE and WT/SL comparisons after inoculation (Figure 7g).

### 2.7. Effects on Differential Gene Expression in Phenylpropanoid Metabolic Pathways

The genes implicated in phenylpropanoid production pathways were specifically analyzed in this study. Nine peroxidase genes were among the 42 differentially expressed genes (DEGs) involved in these metabolic pathways that were found by the results (Figure 8). The highest expression levels were shown by *Zm00014d035941*, which remained consistently high before and after infection in all lines. In contrast to WT, *Zm00014d037955* expressed less in SL lines and more in OE lines throughout the late stages of infection. Six β-glucosidase genes were identified by analysis. During infection times, the majority of these genes were virtually inactive in *SDG102* transgenic lines. Of the five CAD genes found within the phenylpropanoid metabolic pathway, *Zm00014d005093* showed noticeably greater expression in the middle stages of infection in both OE and SL lines than in the early and late stages. Notably, *PNC* gene expression levels in OE lines exceeded those in WT following pathogen infection.

### 2.8. Impact of SDG102 on Corn Yield

The 2024 yield comparison trial demonstrated that the OE line overexpressing *SDG102* exhibited significantly higher traits than the wild-type in ear length, ear diameter, ear weight, number of rows per ear, seed yield, and 100-seed weight, with increases ranging from 4.47% to 18.05%. Under large spot disease conditions, the OE line exhibited a significant 11.46% increase in yield. Conversely, the SL knockdown line showed significantly higher ear diameter, kernel length, and seed yield than the WT, but exhibited significantly reduced ear weight, kernels per row, and kernel width, resulting in a 5.65% yield reduction. These results indicate that *SDG102* overexpression enhances maize resistance to large spot disease while improving yield (Table 1).

### 2.9. Validation of RNA-Seq Results by qRT-PCR Analysis

We randomly selected 12 genes and validated them using quantitative real-time PCR with specific primers (Appendix A) to confirm the accuracy of RNA sequencing data obtained from wild-type and transgenic plants after infection with *S. turcica*. For the majority of the chosen genes, as illustrated in Figure 6, the correctness and dependability of our transcriptome profiling results are confirmed by the excellent consistency seen for the majority of examined genes (Figure 9).

## 3. Discussion

### 3.1. Effects of SDG102 Gene on Agronomic Traits and Physiological–Biochemical Indices in Maize Under S. turcica Stress

Prior research has shown that histone lysine methylation has a variety of functions in plant disease resistance [26], with pathogen infection causing certain histone changes that either stimulate or inhibit gene transcription [27]. According to our ex-pression profiling, *S. turcica* infection dramatically increases the expression of the *SDG102* gene in maize, suggesting that this gene plays a functional role in the plant’s defense against this pathogen (Figure 4b). Histone lysine alterations improved seed germination rates un-der low-temperature stress, according to Qi et al. (2023) who also reported that maize lines overexpressing *SDG102* showed improved cold tolerance when compared to wild-type [28]. In a similar vein, Liu et al. (2021) showed that rice plants overexpressing Os*SDG721* exhibited higher saline-alkali tolerance, whereas loss-of-function mutants showed increased susceptibility to saline-alkali stress [29]. Our research showed that plants overexpressing *SDG102* had greater resistance to *S. turcica*, while lines silenced by *SDG102* showed greater vulnerability to pathogen attack, larger lesion regions, and higher disease indices in the milk stage (Figure 2). Various physiological indicators accumulate and coordinate defense responses against biotic and abiotic stresses, demonstrating the sophisticated systems that plants have evolved to cope with ad-verse environmental conditions [30]. In order to increase resistance and reduce cellular damage after pathogen infection, plants regulate the formation of reactive oxygen species (ROS) and trigger enzymatic and non-enzymatic antioxidant defense systems [31]. Our research showed that in both *SDG102*-overexpressing and silenced transgenic maize materials, the development of disease symptoms (lesion number and size) (Figure 1), ROS buildup, and antioxidant enzyme activities (Figure 3) were significantly impacted by *S. turcica* infection. Simultaneously, transcriptomic data revealed that compared to WT, the genes encoding *ZmCAD1*, *ZmPRX44*, *ZmPRX5*, and *ZmPRX65* were significantly upregulated in the OE line under *S. turcica* stress, while no significant changes were observed in SL. This indicates that overexpression of the *SDG102* gene elevates the transcriptional levels of most genes involved in reactive oxygen species (ROS) production (Appendix A). The results above indicate that overexpression of the *SDG102* gene enhances the expression of genes associated with the “reactive oxygen species production” pathway. This enables OE transgenic plants to accumulate reactive oxygen species more effectively during the early stages of *S. turcica* infection, thereby conferring stronger resistance to leaf spot disease.

### 3.2. Transcriptomic Analysis of SDG102-Mediated Responses to S. turcica Infection

Previous studies have shown that distinct histone modifications regulate key genes associated with seed dormancy, abscisic acid (ABA) and gibberellin (GA) syn-thesis, and signaling pathways [32,33,34]. Dong et al. (2008) proposed that the histone methyltransferase *SDG8* regulates H3K36 methylation of histone-associated genes in Arabidopsis, participating in photosynthesis, metabolism, and energy production [35]. Therefore, it is speculated that overexpression of *SDG102* in maize may catalyze H3K36 methylation of histones associated with resistance-related genes (*PR4*, *WAK-RLK1*, *NPR1*, *PP2C26*) and antioxidant enzyme genes (SOD, POD, PAL), thereby activating transcriptional activity and upregulating expression to positively regulate maize resistance against *S. turcica* infection. (Figure 4c,d). Based on GO and KEGG enrichment analyses, *SDG102* overexpression significantly enhanced the upregulation of differentially expressed genes. Upregulated genes in overexpressing plants were primarily associated with pathways including metabolic function, cellular components, response to stress, plant–pathogen interactions, and plant hormone signaling (Figure 5). This indicates that *SDG102* modulates the response to pathogen stress by stimulating gene expression regulation of target genes.

### 3.3. SDG102 Regulates Expression of Genes Involved in Hormone Signaling and Secondary Metabolite Biosynthesis

MAPK signaling pathway is a primary signal amplifier immediately downstream of pathogen perception, when pattern-recognition receptors (PRRs) detect pathogen-associated molecular patterns (PAMPs), its activation of early defense genes [36]. Second, the MAPK cascade effectively “hands off” the signal to the plant hormone signal transduction pathways. Plant defense mechanisms against pathogens involve a variety of phytohormones, such as salicylic acid (SA), jasmonic acid (JA), brassinosteroids (BR), and cytokinins (CTK) [37]. The Salicylic Acid (SA) pathway is predominantly employed against biotrophic pathogens, leading to Systemic Acquired Resistance (SAR) and the expression of Pathogenesis-Related (PR) proteins. In contrast, the Jasmonic Acid (JA) and Ethylene (ET) pathways are central to defense against necrotrophs and herbivores [38]. On the other hand, to fuel these energy-demanding processes, a profound reprogramming of starch and sucrose metabolism occurs. Defense is a costly endeavor, requiring ATP, reducing power, and carbon skeletons. Upon infection, plants often break down starch into soluble sugars, which serve as a direct energy source Via respiration and as signaling molecules themselves. Finally, the phenylpropanoid biosynthesis pathway represents a key “defense arsenal”, this pathway is highly induced during infection and produces a vast array of defensive compounds to inhibit pathogen growth [39]. In particular, SA and JA are important signaling chemicals that mediate plant immunity, defenses against bio-trophic diseases are predominantly regulated by the SA signaling pathway [40]. Combined with transcriptome sequencing analysis, *SDG102* regulates gene expression related to the “plant hormone signaling pathway.” Following *S. turcica* infestans stress, some genes associated with this pathway were upregulated in the OE strain but downregulated in the SL strain (Figure 6). It also modulates the synthesis of JA and SA hormones and the expression of hormone signaling genes (Figure 7). Concurrently, numerous DEGs participate in the phenylpropanoid metabolism pathway, with higher expression levels of key DEGs in the OE line (Figure 8). This indicates that SDG102 activates multiple metabolic pathway response genes following *S. turcica* infestans stress, thereby influencing maize resistance to *S. turcica* infestans.

Through a thorough transcriptome analysis of genes linked to hypersensitive response, our study offered fresh insights into the epigenetic function of *SDG102* in pathogen defense. Zhou et al. (2005) provided supporting data by showing that ectopic overexpression of histone deacetylase HDA19 increases ERF1 expression and strengthens resistance to Plasmodiophora brassicae [41]. Similarly, Fromm et al. (2014) found that ATX1 loss-of-function mutations change the expression of many antimicrobial genes [42]. According to Mukherjee et al. (2015), *SDG8* depletion results in a substantial drop in H3K36me3 marks at the 3’ ends of gene bodies, which is accompanied by a decrease in the expression of genes that respond to light and carbon [43]. Therefore, we speculate that *SDG102* may mediate reduced H3K36 methylation levels, thereby leading to decreased resistance of the SL line against *S. turcica*. In order to increase pathogen resistance, Lee et al. (2016) looked into how histone methylation controls the manufacture of secondary metabolites [23]. They found that H3K4me and H3K36me alterations can affect the formation of cuticular wax as well as carotenoid biosynthesis. Cazzonelli et al. (2010) have demonstrated that *SDG8* deletion changes the composition of carotenoid, and that *SDG8*-mediated CRTISO expression requires both the open reading frame and the CRTISO (carotenoid isomerase) promoter [44]. *SDG102* may improve plant defense by controlling the generation of antimicrobial secondary metabolites, according to our study, which also found increased expression of phenylpropanoid pathway genes in OE lines (Figure 8). Another factor that probably contributes to the observed differences in disease resistance between OE and SL gene types is this metabolic reprogramming.

## 4. Materials and Methods

### 4.1. Plant Materials

The maize materials used in this study included the wild-type inbred line B73 (WT), a transgenic RNA interference line targeting *SDG102* (AS3), and a *SDG102*-overexpressing line in the B73 genetic background (OE3), the transgenic materials were obtained by our predecessors (Appendix A) [45]. All transgenic lines were confirmed using immunochromatographic test strips, endpoint PCR, and quantitative real-time PCR (qRT-PCR) for transgene presence and expression level [46].

### 4.2. Pathogen Inoculation and Disease Phenotyping

*S. turcica* isolate DB-28a was cultured on PDA medium at 25 °C in complete darkness for 7 days. Mycelial plugs were transferred to fresh PDA plates and incubated under identical conditions. After colony expansion, plates were sealed for 5 days to induce conidiation. Conidia were harvested using sterile distilled water, centrifuged, and resuspended to a final concentration of 1 × 10^6^ spores/mL in 0.01% (*v*/*v*) sterile Tween-20 [2].

Plants were inoculated at the late whorl stage by evenly spraying the spore suspension onto the third and fourth leaves below the ear, with approximately 10 mL per leaf. Inoculations were conducted under overcast conditions to promote infection. Disease progression was assessed at 0, 5, 10, and 20 days post-inoculation (dpi). Leaf samples from inoculated regions were collected from a minimum of 10 plants per time point, flash-frozen in liquid nitrogen, and stored at −80 °C for subsequent analyses. A subset of plants was inoculated using the sorghum grain method to evaluate resistance during the milk stage.

### 4.3. Histopathological Examination and Symptom Quantification

Disease symptoms were imaged at 0, 5, 10, and 20 dpi under consistent lighting conditions. Lesion area was quantified using ImageJ-1.54h (National Institutes of Health, NIH, Bethesda, MD, USA). For microscopic observation, leaf segments (0.5 cm^2^) were fixed in ethanol-acetic acid (1:1, *v*/*v*) for 24 h, cleared in saturated chloral hydrate, and stained with 0.05% aniline blue. Samples were mounted in 50% glycerol and observed under a Nikon Eclipse E100 microscope (Nikon Corporation, Tokyo, Japan) equipped with a DS-Fi3 camera (Nikon Corporation, Tokyo, Japan).

### 4.4. Physiological Characteristics and Hormonal Analyses

Soluble protein content was determined using Bradford assay with BSA as standard. Soluble sugars were quantified by the anthrone–sulfuric acid method [47]. Hydrogen peroxide (H_2_O_2_) and superoxide anion (O_2_^−^) levels were measured [48]. Activities of antioxidant enzymes (SOD, POD, CAT, APX) and PPO were assayed using established spectrophotometric methods. Malondialdehyde (MDA) content was determined Via thiobarbituric acid reaction. Endogenous phytohormones were extracted and profiled using UPLC–MS/MS with multiple reaction monitoring [49].

### 4.5. Transcriptome Sequencing and Analysis

Total RNA was extracted from inoculated leaf samples collected at 0, 5, 10, and 20 dpi using TRIzol reagent. RNA integrity was verified using an Agilent 2100 Bioanalyzer (Agilent Technologies Inc., CA, USA). Libraries were constructed and sequenced on an Illumina HiSeq 2500 platform (Novogene, Sacramento, CA, USA) (150 bp paired-end). Reads were quality-trimmed and aligned to the B73 reference genome (v4). Differentially expressed genes (DEGs) were identified using DESeq2 with FDR < 0.01 [50,51]. Functional enrichment analysis was performed using AgriGO v2 with Fisher’s exact test. Differential expression analysis of two groups (three biological replicates per condition) was performed using the DESeq2 R package (1.20.0). Gene Ontology (GO) and Kyoto Encyclopedia of Genes and Genomes (KEGG) tools (https://www.kegg.jp/) were used to analyze the DEGs.

### 4.6. Quantitative PCR Validation

cDNA was synthesized from DNase-treated RNA using TransScript II Reverse Tran scriptase. qPCR was performed on a QuantStudio 6 Pro system (Thermo Fisher Scientific Inc., San Francisco, CA, USA) using SYBR Green master mix (Thermo Fisher Scientific Inc., CA, USA). The Actin1 gene (GRMZM2G126010) served as the internal control. Relative expression was calculated using the 2^−ΔΔCT^ method [52] with three biological and three technical replicates per sample.

### 4.7. Assessment of Yield-Related Agronomic Traits

In 2024, yield-related agronomic traits and plot yield comparisons were conducted across different materials. Surveys of yield-related agronomic traits and plot yields were performed after ears reached full maturity. For each material, two rows of ears were selected for yield-related agronomic trait measurements. After complete air-drying, ears were threshed to measure traits including kernel length, kernel width, ear weight, ear length, ear diameter, cob diameter, kernel yield, number of rows per ear, kernels per row, and hundred-kernel weight. Plot yields were simultaneously determined.

### 4.8. Statistical Analysis

Data were analyzed using SPSS 26.0 (IBM, USA). Significance between groups was determined by one-way ANOVA followed by Tukey’s HSD test (*p* < 0.05). All values are presented as mean ± standard deviation (SD). Figures were prepared using GraphPad Prism 9.3 and Adobe Illustrator 2022 [48].

## 5. Conclusions

In this study, overexpression of *SDG102* inhibits *S. turcica* spore germination and mycelial growth in plant leaves, enhances antioxidant enzyme activity and reactive oxygen species scavenging capacity during the early stages of *S. turcica* infection, and reduces oxidative damage in maize. Conversely, silencing *SDG102* increases susceptibility to fungal infection. Transcriptome sequencing analysis identified 1546 and 1837 differentially expressed genes in OE and SL in response to *S. turcica* infection. These differentially expressed genes primarily functioned in pathways including plant–pathogen interactions, plant hormone signaling, biosynthesis, transport, and catabolism of secondary metabolites, as well as starch and sucrose metabolism. Overexpression of *SDG102* activated more differentially expressed genes involved in pathogen defense mechanisms, while silencing *SDG102* had the opposite effect. Furthermore, *SDG102* regulates the synthesis of the hormones JA and SA, as well as the transcription of their signaling pathway genes, thereby influencing maize resistance to large spot disease. Under conditions of corn leaf spot disease, compared to the wild-type, the OE variant showed an 11.46% increase in yield, while the SL variant exhibited a 5.65% decrease in yield (Appendix A). Taken together, these data provide novel and valuable insights into understanding the mechanisms underlying maize resistance to *S. turcica* in response to *SDG102*.

## Figures and Tables

**Figure 1 plants-14-03463-f001:**
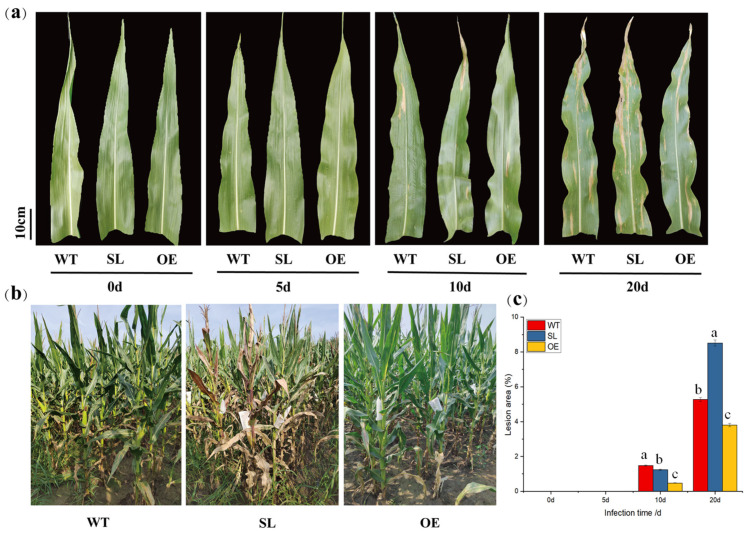
Response of *SDG102* transgenic maize to *S. turcica* infection, leaf diseases at different time points after inoculation with *S. turcica*: (**a**) Leaf disease observed in wild-type and *SDG102* transgenic plants at various infection time points; (**b**) Field disease incidence of NCLB during the milking period of the test materials; (**c**) Comparative analysis of plaque area across different infection time points. Different letters indicate significant differences between materials at the same time point (*p* < 0.05).

**Figure 2 plants-14-03463-f002:**
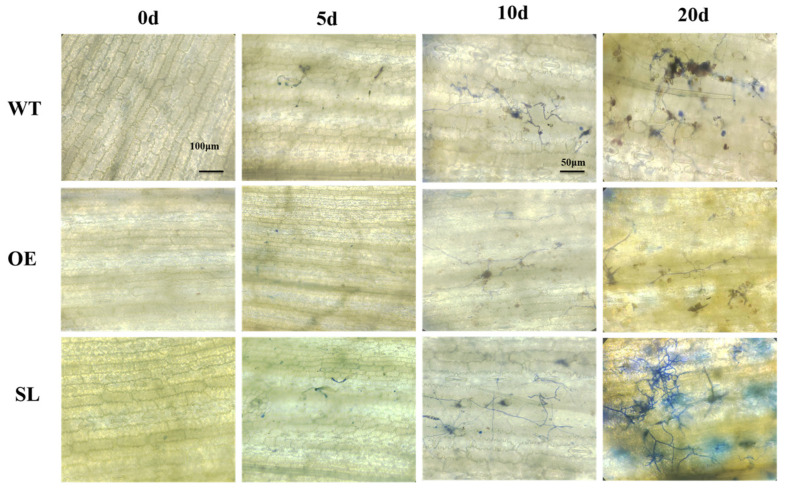
Microscopic view of inoculated *S. turcica*.

**Figure 3 plants-14-03463-f003:**
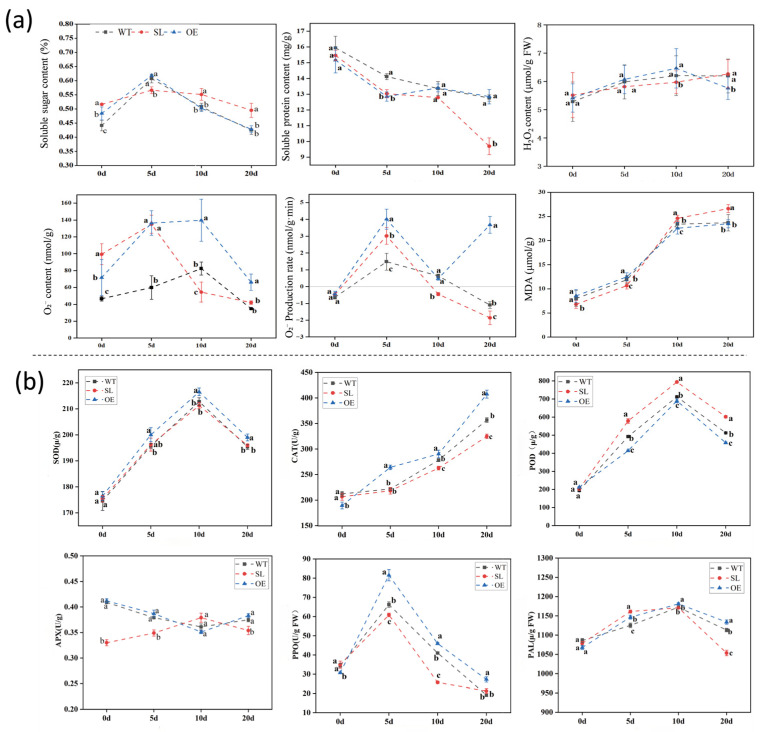
(**a**) Alterations in the concentrations of soluble compounds and peroxide derivatives in leaves following infection by *S. turcica*. (**b**) Trends in antioxidant enzyme activities after *S. turcica* infection. Using one-way ANOVA and Duncan’s tests to compare the data, different lowercase letters indicate significant differences (*p* < 0.05).

**Figure 4 plants-14-03463-f004:**
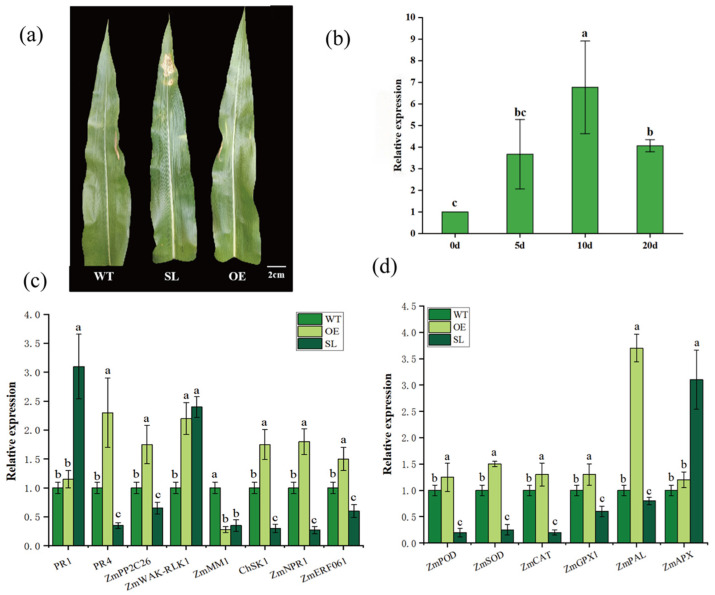
Expression analysis of pathogen stress response genes in *SDG102* transgenic maize: (**a**) Leaf pathogenic symbols at 10 d of inoculation with *S. turcica*; (**b**) Expression pattern of *SDG102* gene in maize after inoculation with *S. turcica*; (**c**) Relative expression of genes related to resistance to NCBL; (**d**) relative expression of genes encoding antioxidant enzymes. Note: One-way ANOVA was performed using Tukey’s honest significant difference (HSD) test, and different letters indicate statistically significant differences (*p* < 0.05).

**Figure 5 plants-14-03463-f005:**
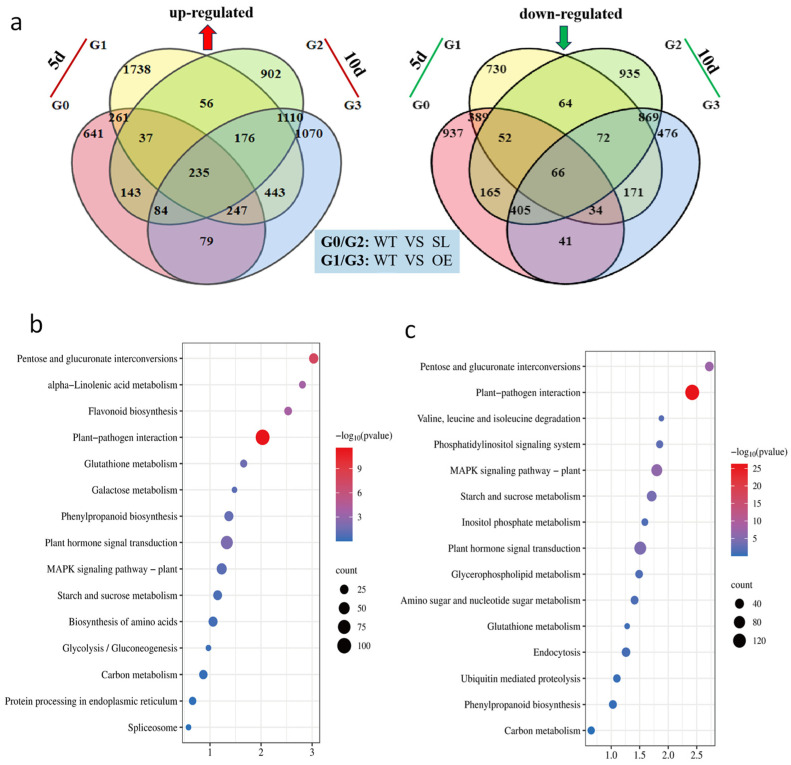
Transcriptome differential gene and enrichment analysis: (**a**), Venn map of OE and SL differentially expressed genes under *S. turcica* infestation conditions; (**b**), KEGG enrichment map of WT-10d vs. OE-10d; (**c**) KEGG enrichment map of WT-10d vs. SL-10d. Note: Venn diagram shows DEGs expression profiles between OE, SL and WT under pathogen stress (q < 0.01, fold change (FC) > 2).

**Figure 6 plants-14-03463-f006:**
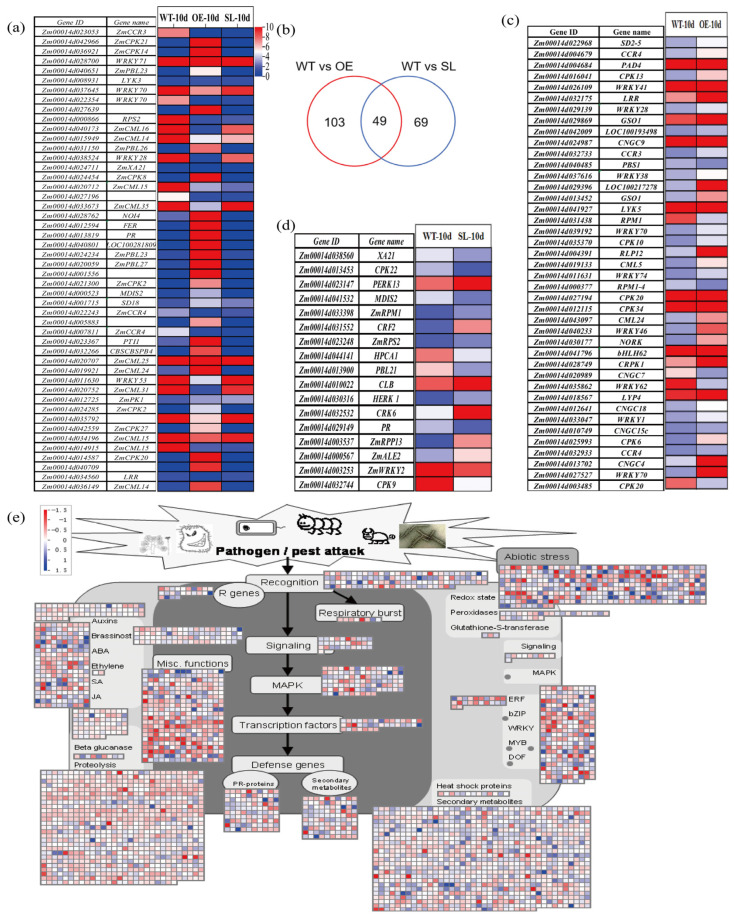
Heatmap of expression level of genes related to “plant–pathogen interaction pathway”: (**a**) Venn diagram of differential genes in the “plant–pathogen interaction pathway”; (**b**) Expression analysis of genes specific to OE plants in the “plant–pathogen interaction pathway”. (**c**) Expression analysis of genes specific to SL plants in the “plant–pathogen interaction pathway”; (**d**) Expression analysis of genes common to the “plant–pathogen interaction pathway”; (**e**) Heat map of plant–pathogen interactions in OE response to *S. turcica* infection.

**Figure 7 plants-14-03463-f007:**
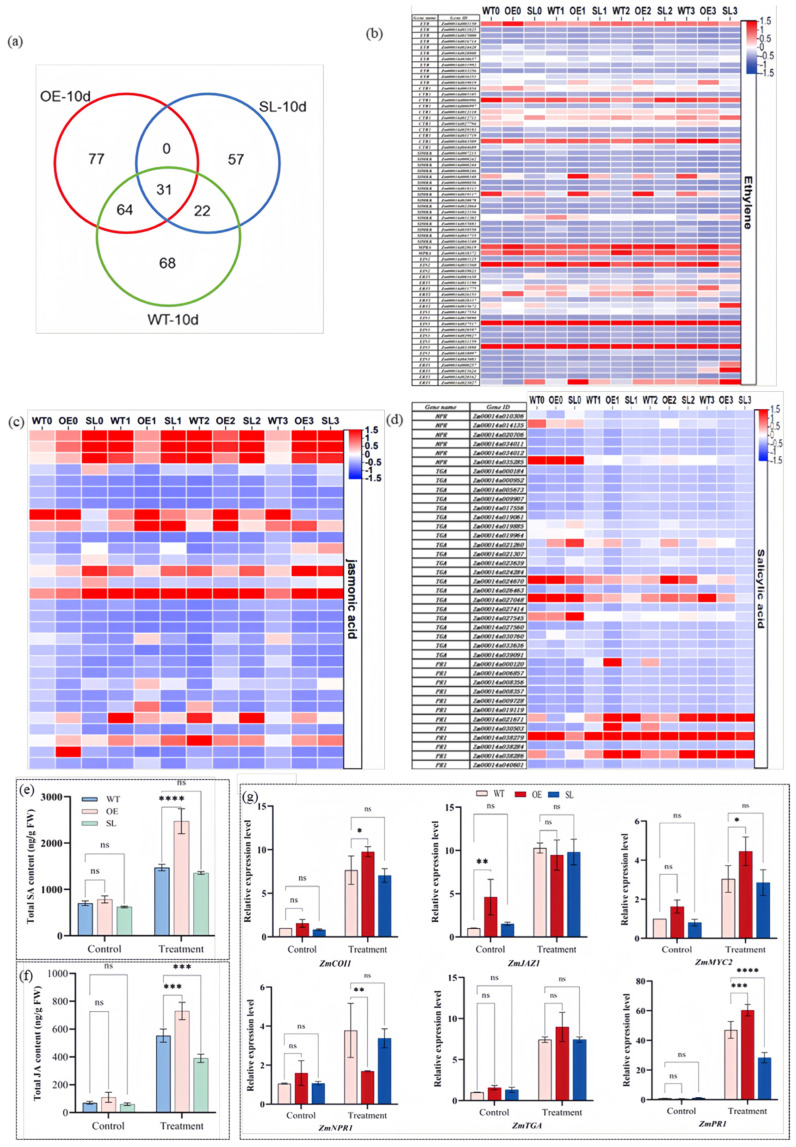
Analysis of hormone signaling related to OE and SL transgenic plants: (**a**) Venn diagram illustrating the genes involved in the ‘plant hormone signal transduction pathway’; (**b**) Expression of genes involved in ET signal transduction pathway; (**c**) Expression analysis of genes involved in JA signal transduction pathway; (**d**) SA signal transduction pathway; (**e**) Contents of salicylic acid (SA) in the leaves of *S. turcica* under infection conditions; (**f**) Contents of jasmonic acid (JA) in the leaves of *S. turcica* under infection conditions; (**g**) Analysis of gene expression related to JA and SA signaling pathways in *SDG102* transgenic maize plants following inoculation with *S. turcica*. Note: The significant difference was evaluated by the Student’s *t*-test. * *p* < 0.05, ** *p* <0.01, *** *p* <0.001, **** *p* <0.0001, “ns” indicates no significant difference.

**Figure 8 plants-14-03463-f008:**
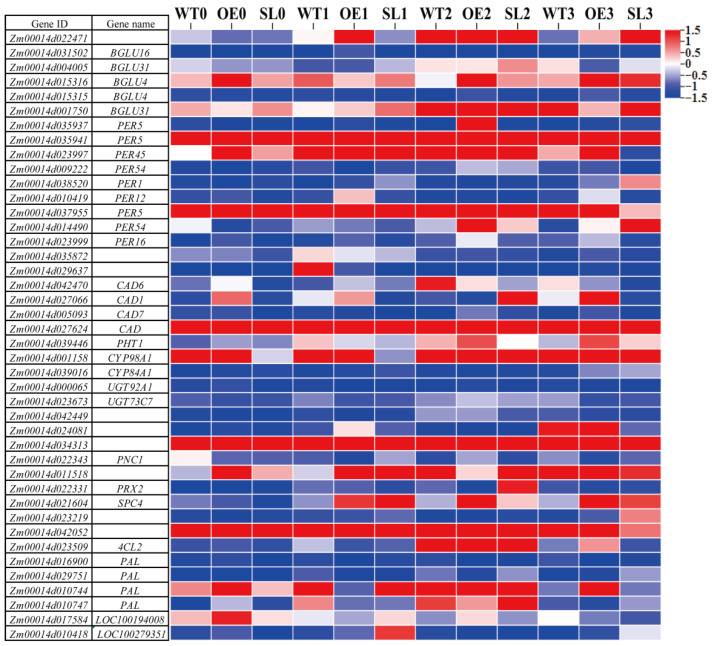
Heat map analysis of genes associated with the phenylpropane metabolic pathway.

**Figure 9 plants-14-03463-f009:**
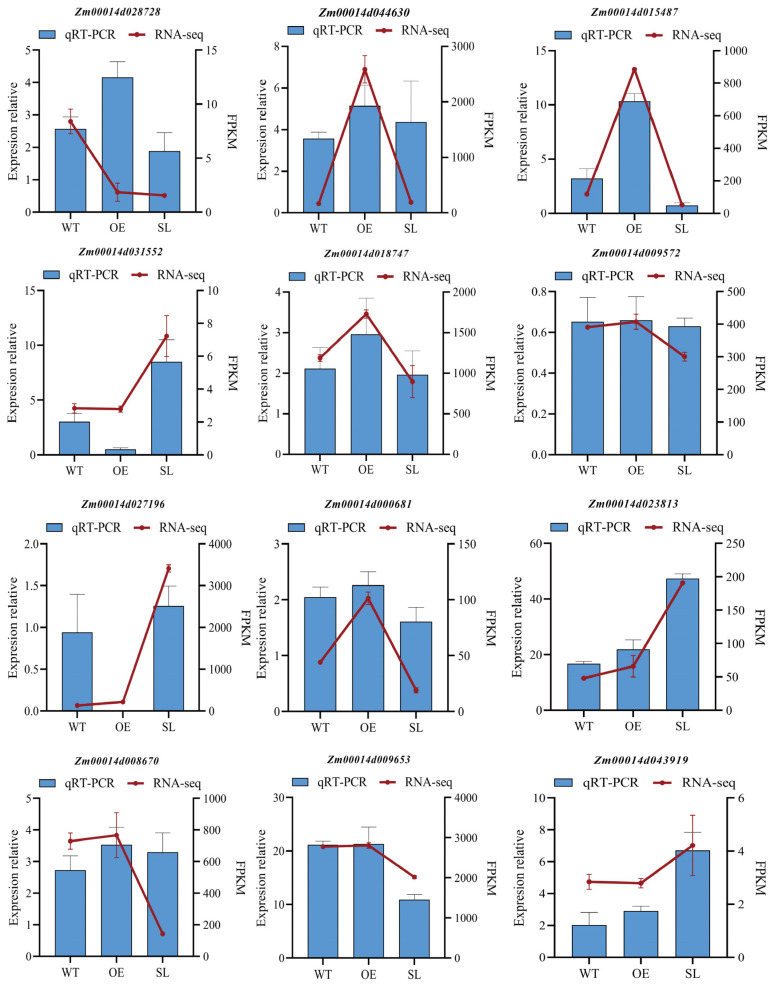
The results of RNA-seq and qRT-PCR were compared. Relative expression levels of 12 genes in three maize lines. Expression of *SDG102* measured by qRT-PCR (left bars) and RNA-seq (right bars) in wild-type, *SDG102*-overexpression, and *SDG102*-silenced lines. Data are presented as mean ± standard deviation (SD). The arrow connecting the bars signifies a comparison between the two technical platforms (qRT-PCR vs. RNA-seq) for the same gene in the same line.

**Table 1 plants-14-03463-t001:** Survey of yield-related agronomic traits among different materials.

Name	Length of Spike (cm)	Diameter of Spike (cm)	Weight of Spike (g)	Length of Grain (cm)	Width of Grain (cm)	Number of Panicle Rows	Number of Grains in Row	Diameter of Cob (cm)	Yield of Seed (%)	100-Kernel Weight (g)	Plot Yield (kg)	Yield/667 m^2^
WT	15.76 ± 0.43 b	3.93 ± 0.09 c	100.79 ± 1.67 c	1.00 ± 0.06 b	0.76 ± 0.05 a	15.30 ± 0.94 b	24.5 ± 0.50 a	2.75 ± 0.27 a	75.70 ± 1.44 b	25.60 ± 0.10 b	12.21 ± 1.14 b	298.31 ± 27.85 b
OE	17.22 ± 0.69 a	4.30 ± 0.08 a	118.98 ± 2.80 b	1.12 ± 0.06 b	0.82 ± 0.07 a	16.00 a	23.75 ± 0.83 a	2.72 ± 0.15 a	79.09 ± 0.20 a	27.00 ± 0.71 a	13.61 ± 1.98 a	332.52 ± 48.37 a
SL	15.08 ± 0.94 b	4.20 ± 0.05 b	90.28 ± 2.60 a	1.17 ± 0.07 a	0.60 ± 0.10 b	15.00 ± 0.92 b	23.00 ± 1.58 b	2.97 ± 0.41 a	79.01 ± 0.42 a	25.16 ± 0.47 b	11.52 ± 2.86 b	281.45 ± 69.87 b

Note: Values are measured mean ± standard deviation, one-way ANOVA was performed using Tukey’s honest significant difference (HSD) test, and different letters indicate statistically significant differences (*p* < 0.05).

## Data Availability

These sequence data have been submitted to the GeneBank database under Bioproject PRJNA1332421.

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
