# Peer review of "Functional Analysis of Maize SDG102 Gene in Response to Setosphaeria turcica"

_plants, 2025, doi:10.3390/plants14223463_

Round 1

Reviewer 1 Report

Comments and Suggestions for Authors

The manuscript by Qi et al investigates the function of SDG102, a histone H3K36 methyltransferase homolog, in maize resistance to Setosphaeria turcica, the causal agent of Northern Corn Leaf Blight (NCLB). The authors combine phenotypic evaluation, biochemical assays, and transcriptomic analyses of wild-type (WT), SDG102-overexpressing (OE), and silenced (SL) maize lines to explore the gene’s role in defense responses and yield performance. The work is valuable from a translational breeding perspective but before publication, some points should be addressed:

  1. The core hypothesis is that SDG102 functions as an H3K36 methyltransferase regulating defense-related transcription. However, this remains inferred from sequence homology and gene expression. For this reason, the Authors should include test the hypothesis by ChIP-qPCR or western blot assays or discuss why these procedures are not necessary. 
  2. Disease resistance assessment is based on lesion area and disease index. Please, include pathogen biomass quantification (e.g., S. turcica-specific qPCR) or microscopic quantification of fungal structures to confirm phenotypic observations. In alternative, please clarify why these experiment are not necessary.
  3. Some figures and tables lack explicit mention of biological vs. technical replicates and the type of error bars (SD or SE).
  4. Please ensure all ANOVA and post hoc test details (n, p-values) are clearly provided in figure legends.
  5. A schematic figure summarizing the proposed SDG102–mediated defense network (linking histone methylation, hormone signaling, ROS metabolism, and yield) would enhance interpretability.

Minor comments:

  1. The reference list is thorough but should follow consistent formatting (check journal style for punctuation and italics).
  2. Consider citing recent reviews on epigenetic regulation of maize immunity (2023–2025) if available.

Comments on the Quality of English Language

TheEnglish  requires professional or native-level editing for grammar and  style.The tone could be made more concise in discussion and conclusion.

Reviewer 2 Report

Comments and Suggestions for Authors

The authors of the manuscript ‘Functional analysis of maize (Zea mays L.) SDG102 gene in response to Setosphaeria turcica’ reported that overexpression of SDG102 significantly inhibited pathogen spore germination and hyphal growth, and increased the activity of antioxidant enzymes and Zea mays ability to scavenge reactive oxygen species before Setosphaeria turcica infection.

Abstract: Please expand the abbreviation at the first use and then consistently use the abbreviation afterward. For example: L16: SET domain gene 102 (SDG 102) and L22, 24, and 25: Delete (OE), (SL), and (WT).

L19: ‘S. turcica’ Please italicize the species name throughout the text.

Keywords: Please avoid using keywords from the title.

Introduction: Please expand the abbreviation at the first use and then consistently use the abbreviation afterward.

L44: ‘Ht1’ Please italicize the gene name throughout the text.

L47: Please indicate all resistance genes identified in maize.

L77: ‘Northern corn leaf blight (NCLB), a destructive foliar disease caused by S. turcica’ Please try to prevent repetition.

Results:

L101: genotype instead of gene types.

L108: Figure 1c. Please specify the mean separations.

L125: Please improve the quality of Figure 2.

L155: Figure 3. Please indicate the statistical differences (mean separations).

L176-180: Please correct it. Different ‘capital letters’ indicate that different materials at the same time or the same material at different times are significantly different (P < 0.01). There are no capital letters in Figure 4.

L249: Please improve the quality of Figure 7a, e and f.

L304: ‘specific test and adjustment used should be 304 stated in Methods).’ Please delete it.

Discussion: Can be improved (L379-381).

Materials and methods: Indicate the biological and technical replicates.

L397-399: Supporting Figures required.

L477: Supplementary Materials: Missing, please provide the files and include the details.

Comments on the Quality of English Language

Moderate editing is required.

Reviewer 3 Report

Comments and Suggestions for Authors

Title: Functional analysis of maize (Zea mays L.) SDG102 gene in response to Setosphaeria turcica

This manuscript explores the role of SDG102, a histone H3K36 methyltransferase gene, in maize defense against Setosphaeria turcica, the causative agent of northern corn leaf blight (NCLB). Using wild-type (WT), overexpression (OE), and silencing (SL) lines of maize (B73 background), the study combines phenotypic, physiological, and transcriptomic analyses to evaluate disease resistance. The authors demonstrate that SDG102 overexpression enhances antioxidant activity, regulates salicylic acid (SA) and jasmonic acid (JA) signaling, and increases resistance to S. turcica, while silencing reduces resistance. The RNA-seq data reveal significant DEGs in plant-pathogen interaction and hormone signaling pathways, providing evidence that SDG102 positively regulates epigenetic defense responses and improves yield under stress conditions.

Major comments:

1. The study provides meaningful insight into epigenetic regulation of disease resistance in maize. However, the novelty could be better emphasized by clarifying how SDG102’s function differs from previously characterized SDG homologs in Arabidopsis and rice.

2. While transcriptomic and physiological data support the proposed role of SDG102, functional assays confirming its methyltransferase activity and histone modification targets (e.g., H3K36me2/3 quantification) would significantly strengthen the mechanistic conclusions.

3. The manuscript would benefit from clearer reporting of biological vs. technical replicates, sample sizes, and statistical tests for each figure. Error bars, significance levels, and number of replicates should be consistently indicated.

4. The RNA-seq section identifies several DEGs and enriched pathways, but the discussion could link these to specific downstream defense mechanisms (e.g., PR proteins, WRKY factors, ROS detoxification). Including a summary figure or schematic would enhance clarity.

5. The manuscript is generally well written but requires careful language editing to improve readability and grammatical consistency. Some sentences are overly long or repetitive, particularly in the Results and Discussion sections.

Minor comments:

6. Define all abbreviations at first mention (e.g., OE, SL, DEGs, dpi).

7. Figures should include clearer legends specifying sample numbers and significance levels.

8. Ensure consistency in gene name formatting (ZmPR1, ZmPAL, SDG102).

9. Supplementary material references (Tables S2–S5) should be explicitly cited within the text.

10. Revise the abstract to highlight novelty and key findings concisely (e.g., yield improvement under pathogen stress).

The study is scientifically sound and offers valuable insight into epigenetic regulation of maize defense. However, before acceptance, the authors should provide additional functional validation, improve statistical transparency, and refine the discussion and language for better clarity and impact. Once revised, this work could make a solid contribution to Plantsand the field of plant molecular epigenetics.

Round 2

Reviewer 1 Report

Comments and Suggestions for Authors

The Authors have addressed all comments.